# Dealing with Access to Spirometry in Africa: A Commentary on Challenges and Solutions

**DOI:** 10.3390/ijerph16010062

**Published:** 2018-12-27

**Authors:** Refiloe Masekela, Lindsay Zurba, Diane Gray

**Affiliations:** 1Department of Paediatrics and Child Health, Nelson R Mandela School of Clinical Medicine, College of Health Sciences, University of KwaZulu-Natal, Durban 4013, South Africa; 2Education for Health Africa, Durban 4302, South Africa; linds@icon.co.za; 3Department of Paediatrics and Child Health and MRC Unit on Child and Adolescent Health, University of Cape Town, Cape Town 7700, South Africa; diane.gray@uct.ac.za

**Keywords:** spirometry, Africa, lung disease

## Abstract

Spirometry is an important tool in the surveillance, epidemiology, diagnosis, and management of respiratory disease, yet its accessibility is currently limited in Africa where the burden of respiratory diseases is amongst the highest globally. The reasons for limited access to spirometry in Africa include poor access to training and skilled technicians, limited availability of equipment, consumables, and technical support, and lack of human and financial resources. The Pan African Thoracic Society, working together with regional African thoracic societies and key research initiatives in Africa, have made progress in training and education, but a lot of work is still needed to meet the challenges faced. Accurately defining these challenges of access to high quality spirometry, development of local, standardised, and context-specific training and quality assurance tools; development of appropriate reference standards and innovative approaches to addressing the challenges of access to equipment, consumables and technical support are needed. Training and research collaborations that include regional thoracic societies, health system leaders, the Pan African Thoracic Society and international role players in the field are key to maximising available intellectual and financial resources. Hence ensuring that access to high quality spirometry measures that are used effectively in tackling the burden of respiratory disease in Africa.

## 1. Introduction

Spirometry is an objective tool for the diagnosis, severity assessment, management, risk factor categorization, and follow-up of patients with chronic lung disease. In the context of occupational health, it can be utilized for pre-placement screening and follow-up of workers to monitor and minimize risk for individuals in high-risk occupations. However, despite its key role in optimizing respiratory health, access to spirometry is limited in many low–middle income countries (LMICs). In LMICs often the challenge with delivery of healthcare is that services are skewed to treating of acute episodes and not for chronic disease prevention, management, and follow-up. This is despite growing evidence of the burden of non-communicable diseases (NCDs) in LMICs, of which chronic respiratory disease (CRD) is a leading contributor. This article aims to describe the current challenges in access to spirometry in many African countries, present some of the work that is currently being done by the Pan African Thoracic Society (PATS) to improve access and to identify important areas for further research and action.

## 2. Methodology

A PubMed search was undertaken to identify published papers on the use of spirometry and spirometry training initiatives in Africa. Search terms included “spirometry” OR “lung function” AND “Africa” AND “training” OR “guidelines”. The search was limited to publication years 1970 to 2018 and English language. An internet search, using search engine Google Chrome with search terms “spirometry training” OR “lung function training” AND “Africa” or “South Africa”, was also undertaken to look for unpublished information on spirometry training courses. In addition, respiratory physicians and allied staff working in respiratory medicine and research in Africa were contacted through the PATS database and asked for information regarding their centre’s access to, use of, and challenges with spirometry for clinical or research purposes.

## 3. NCDs and CRD in Africa: Burden of Disease

There is an increasing prevalence of non-communicable diseases in Africa and this is negating the benefits reaped from success in reductions in infectious diseases [1]. Of these, NCDs account for around 63% of deaths, which include the major four: cardiovascular diseases, diabetes, cancer, and chronic respiratory diseases [2,3]. The impact of NCDs is far-reaching as they threaten economies of many LMIC countries, placing high demands on health systems, particularly where populations are aging [4].

Of the NCDs, chronic respiratory diseases have increased with the Global Burden of Disease Report 2015, showing that 4 million deaths are due to CRDs and of these 80–90% occur in LMICs [5]. Despite the high burden of CRDs and high mortality rates, the WHO Global Monitoring Framework did not set any targets for CRDs in 2011 [6].

Asthma rates have been increasing in the Africa region with one systematic review predicting asthma prevalence rates to have tripled in the last three decades with estimated numbers of 41 million asthmatics in Africa in 1990 and 119 million by 2010 [7]. Worryingly, with increasing urbanization, the rates of CRDs are also increasingly linked to indoor and outdoor pollution, poor nutrition, and infections like tuberculosis (TB) and human immunodeficiency virus (HIV) which have devastating consequences and increase the prevalence of chronic obstructive pulmonary disease. With this growing burden of CRDs, there is clearly a need for access to appropriate diagnostic, management, and research tools.

## 4. Current Access to Spirometry: Gaps in the System

In the context of Africa, the current limitations for access to spirometry are multi-factorial. The challenges include limited training and testing guidelines, limited availability of equipment and technical support, lack of human and financial resources and government support, and paucity of local relevant research.

### 4.1. Technologist Training and Guidelines

Many African countries do not have access to pulmonary function laboratories and even in tertiary centres where laboratories exist; there are limitations in terms of the tests that can be performed as well as a dearth of skilled technologist to perform these specialized tests. In one small study in Nigeria assessing the practice of respiratory specialists and fellows, only 34% of the cohort had the ability to perform basic spirometry, while none had capacity to perform challenge testing [8].

There are few African spirometry-training programs available apart from formal programs at institutions of higher education which train respiratory technologists. This makes standardization of spirometry practice by non-respiratory technologists challenging. In addition, the literature search on published and unpublished information on spirometry training in Africa found that spirometry training is only currently offered in South Africa by a handful of Universities and Technical Colleges and another handful of individuals teaching courses of their own making [9,10,11]. No formal courses were found in any other African country. There are also no spirometry training accreditation or endorsement guidelines in place in South Africa or Africa for those offering training, as far as we can ascertain. This makes it unclear which personnel are accredited or allowed to perform spirometry and limits standardisation, quality assurance and continued professional development.

Local guidelines on the performance of spirometry are necessary to take into account the local context and disease spectrum in a population. From our literature review we found that only one African country (South Africa) had published local guidelines for both adults and children [12,13,14].

### 4.2. Equipment and Technical Support

Access to spirometry equipment is reported as poor in the few published African papers [8,15,16]. Kibirige et al. published results of data collected from 22 public hospitals, 23 private hospitals and 85 private pharmacies in Uganda with results showing that spirometry was available in only 13.6% of the private hospitals and 34.8% of public hospitals [15].

In 2013, 40 Nigerian Thoracic Society members responded to a Pan African Thoracic Society Methods in Epidemiological and Operational Research (PATS MECOR) administered questionnaire on the process and costs of respiratory medicine training and facility, equipment, and supply capacities at the institutions they represented [8]. Only 34% of respondents had access to full spirometry on patients. Nwosu et al. in Enugu, Nigeria, conducted a follow-up study to assess the current pattern of utilisation of spirometry in their centre by comparing data from 2010 to 2016. They found spirometry usage to be on the rise, but still far below ideal [16].

Moreover, local spirometry equipment service providers are available in only a few African countries. This means that equipment costs escalate, as they must be shipped from other countries. There is poor access to on-site technical support, scarce equipment calibration facilities and long delays in access to consumables [11]. In some instances, more costly consumables are required, for example, additional bacterial filters due to the high burden of infectious diseases, including tuberculosis (TB). In some cases where interested physicians have purchased lower-cost spirometers on-line without the knowledge of the American Thoracic Society (ATS) and European Respiratory Society (ERS) technical specifications for spirometry, they have been challenged with software insufficiencies that limit their use [11]. The hurdle of starting to practice spirometry without spirometry training or local technical support can result in sub-standard spirometry practice or the equipment being set aside.

### 4.3. Human Resources

Spirometry access does not only depend on the equipment but also on physicians requesting and accurately interpreting spirometry results. There are a number of disparities in African countries in terms of access to specialist care with figures from the African regions demonstrating very low numbers. The number of physicians per 1000 population (as per WHO data) varies from 0.019 per 1000 in Malawi and 0.198 per 1000 in Kenya to 0.408 per 1000 in Nigeria, and 0.779 per 1000 in South Africa [17]. These figures are well below those in high-income countries, which vary between 2 and 4 per 1000 population [17]. For the paediatric population, the WHO recommends 10 paediatricians per 100 000 population, with numbers in high income countries ranging from 11 to 86 in the United Kingdom and Germany, respectively [18]. Data from African countries range between 0.03 and 0.8 per 1000 000. In terms of access to pulmonologists one study in a paediatric population showed that many African countries have no paediatric pulmonologist with only 11 paediatric pulmonologists trained over a 10-year period [19]. In a literature review of diagnosis of COPD in Africa, even when specialists were available, the majority of physicians did not use spirometry for diagnosis [20].

### 4.4. Financial Resources

Healthcare budgets in LMICs have competing interests, and the majority of countries having to prioritize acute emergency care with management of chronic diseases. In the state-funded sector, access to specialized testing and equipment is limited and there is inequity in terms of access to care in the state-funded system when compared to private or insurance based healthcare. A Ugandan study reported spirometry to be unaffordable to the majority of Ugandan patients with asthma and COPD [15]. The vast majority of the population in LMICs is reliant on the state-sector based healthcare system; which in most cases cannot provide spirometry nor personnel to perform testing. Other challenges include a stable electricity supply and access to electricity in more rural facilities. These issues also dictate the type of equipment which can be utilized for testing.

### 4.5. Research

Interpretation of spirometry is based on appropriate references for the individuals’ age, gender, height and ethnicity compared to the local population. This requires the collection from large numbers of healthy people form the population. With the exception of North Africa, this has not been robustly collected in the majority of African populations. Hence, in many African settings, spirometry interpretation has relied on reference data from non-African populations, often using weakly based estimated ethnic correction factors, which can lead to over or under diagnosis of lung disease. The Global Lung Initiative 2012 (GLI_2012_) has more recently provided the most robust and physiologically sound reference equations for multiple ethnicities (Caucasian, African American, North and South East Asian) for use in spirometry. [21] The limitation of GLI_2012_ was the lack of data from African countries with the recommendation that in ethnicities where there is no local data the “Other” reference be utilized. The appropriateness of use of the African-American or Caucasian reference equations in African subjects has yielded conflicting results in studies in Africa, highlighting the importance of locally collected healthy reference data [22,23,24].

## 5. Current Spirometry Training Initiatives in Africa for Clinical and Research Purposes: Pan African Thoracic Society

PATS and affiliated national thoracic societies have made a start at addressing some of these challenges including spirometry training with free access to educational materials, spirometry quality assurance and overreading workshops, a spirometry train-the-trainer programme, freely available spirometry standard operating procedures, and equipment loan for African researchers.

### 5.1. PATS Certificate of Competence in Foundational Spirometry

One of the specific aims of PATS is the promotion of education and training initiatives to strengthen respiratory health across Africa [25]. In line with this, in 2017, PATS developed an international standard foundational spirometry training programme with funding support from the Lung Health in Africa across the life course (LuLi), the National Institute for Health Research (NIHR) Global Health Research, and the European Respiratory Society (ERS). The course has been developed with the primary purpose of increasing knowledge, practice, awareness and access to high quality spirometry in African workplaces and research units.

A blended learning approach is used with a particular focus on our unique African settings and challenges. The full course spirometry training takes a minimum of 2 months to complete and has 3 stages which include distance study, three days of face-to-face training and a portfolio of evidence workbook. The training materials are available for free online at http://panafricanthoracic.org/. All learning materials are available in English, French and Armenian. Future plans include translation into Portuguese. Facilitation is in English with the courses arranged in host-countries for groups. 

To date 12 courses have been completed in 7 countries. This has included 192 students, including nurses, doctors, physiotherapists and research assistants. The majority of students, 89%, have passed the courses, with currently 32% already having successfully completed their portfolios. The success of these programs has included funding for course development and human resources, students access to the updated course material prior to the face-face-training, reliable communication between sites and highly skilled program manager, identification of loyal spirometry champions to ensure ongoing local support and ongoing post-training support and mentorship. Challenges have included funding for trained teaching staff to travel, limited language of teaching materials, lack of availability of equipment to complete portfolios, outdated equipment software to allow appropriate reporting and critically short supply of local technical support.

### 5.2. Spirometry Quality Assurance and Over-Reading Workshop

Spirometry data is pivotal to assessing primary or secondary outcomes in most respiratory trials, but the methodology for data acquisition and data collection is rarely published. In many cases, a significant portion of spirometry data is of inadequate or of questionable quality. These quality issues lead to increased data variability and undermine the validity of the results of a study and are hence crucial.

Through a strong commitment to the highest quality of spirometry in African research the PATS spirometry working group have developed a Quality Assurance and Over-reading Certificate of Attendance training, developed in 2018, which is intended for African respiratory researchers. This aims to establish new foundations for standardised, original and innovative quality assurance processes for research sites in Africa. This 1-day workshop augments the foundational spirometry training programme with spirometry data management, quality assurance and over-reading principles and practice that will increase the quality of the spirometry results for data analysis. The course is planned to be freely available online through the PATS website.

### 5.3. Training Spirometry Trainers

Training opportunities are offered for individuals who would like access to further training to become future PATS spirometry trainers and spirometry champions in host countries. PATS has successfully supported 5 certified PATS spirometry trainers in Malawi, Kenya, Nigeria, and Uganda. There are additionally spirometry facilitators in training from Nigeria, Tanzania, Cameroon, Ghana, and Malawi. 

### 5.4. Spirometry Equipment Loan 

PATS has a limited number of spirometers, donated by the ERS, available for loan to African respiratory researchers. Preference is given to PATS MECOR students [25]. Requests for loan equipment are considered by the PATS spirometry committee on submission of a formal request in writing, submission of the proposal for research and funding statements. 

## 6. Future Areas for Research and Action

Given the many unique challenges described that are faced in Africa and many other LMIC settings on other continents, research that accurately quantifies the challenges to access of spirometry, assesses the optimal use of spirometry in context, assesses innovative approaches to addressing need, and provides high quality research endpoints for clinical research are particularly important. This data is urgently needed to inform African-driven innovation and solutions that improve the respiratory health across the life course.

A number of important research areas that require systematic collection of good quality data and for which spirometry access is key include:Prevalence and incident data of NCDs and CRDs in African countries. This is currently lacking with many estimates extrapolated from data collected from few high and low income countries, hence epidemiological studies estimating the true burden of CRDs across the continent are needed [26]. Some projects such as the Global Asthma Network (GAN) [27] and Burden of Obstructive Lung Disease (BOLD) initiatives that have gone some way to proving this data [28], but these are not yet representative of the entire continent and most specifically the lowest income settings. Spirometry plays an important role in the diagnosis of CRD and hence for accurate measurement of disease burden.Reference ranges for spirometry based on population specific healthy data are needed. The Global Lung Initiative (GLI) has developed a robust all age multi-ethnic reference equation, that does not currently include African data [21]. There have been a number of recently published data from the African region which is progress [22,23,24]. However, broadly representative prospectively collected African data and collaborative research is needed to inform appropriate regional African equations that can inform testing standards and interpretation consistency.Accurate assessment of the type and extent of the challenges to spirometry access: specifically details of equipment used, problems with access to consumables and technical support, and safety, particularly in settings with high infectious disease burden.Assessment of efficacy and sustainability of spirometry training programs; and the impact of improved spirometry access.Establishing clinical utility of spirometry testing in CRDs in Africa. Notably, the impact of screening and early diagnosis of CRD, monitoring of lung function in the management of CRD and disease outcomes still need to be established.Innovative, African-led solutions to the challenges of spirometry access. These may include locally produced, affordable equipment; development of local technical support capacity; innovative approaches to appropriate equipment calibration; optimising infection control in an affordable way and state of the art data management and sharing tools.

## 7. Conclusions

Spirometry is an important tool in the surveillance, diagnosis, and management of respiratory disease, yet its accessibility is currently limited in Africa where the burden of respiratory diseases is amongst the highest globally. The reasons for limited access to spirometry include poor access to training and skilled technicians; limited availability of equipment, consumables and technical support and lack of human and financial resources. Accurately defining the challenges of access to spirometry, development of standardised and context-specific training and quality assurance tools, development of appropriate reference standards, and innovative approaches to addressing the challenges of access are needed. Training and research collaborations that include regional thoracic societies, the Pan African society and international role players in the field are key to maximising available intellectual and financial resources and ensuring that access to high quality spirometric measures become widely available and used effectively in tackling the burden of respiratory disease across the continent.

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
