# Peer review of "Dealing with Access to Spirometry in Africa: A Commentary on Challenges and Solutions"

_ijerph, 2018, doi:10.3390/ijerph16010062_

Round 1

Reviewer 1 Report

I have only one major comments and other minor comments: Major comment: The organization of the paper is somewhat difficult to follow and makes the reading not fluid. There are some overlapping of subjects. In the abstract authors have well identify the key subjects of the limited access to spirometry in Africa. Therefore, the reader expected to recognize the reports on those key problems identified. Unfortunately, I had difficulties to follow. I think that themes might be grouped differently to make the reading more fluent. Example of grouped thematic: - One paragraph on lack of human resources : - Words on training - Technical support - Words on physicians and respiratory specialists - Etc. - One paragraph on lack of financial resources - Budget - - One paragraph on lack of material resources - One paragraph on lack of standards or guidelines: - research - Equations - Etc. - Minor comments First: It is a very relevant paper; unfortunately, the authors did not precise if the paper is a literature review? A report of expert consultation or any other type of paper. They did not give an objective of their manuscript in the introduction paragraph. Second: I think that authors might say one word about the cost of spirometry in Africa. In the paragraph on lack of government support, they can also say a few thing about how people pay medical service (personal payment? Health insurance plan? Etc.) Third: Authors may to define the scope of the study in the term of geographic area. To the best of my knowledge, I think that spirometry is more or less developed in North Africa and that the manuscript has detailed situation in sub-Saharan Africa.

Author Response

Response to reviewer 1:

Thank you for your helpful comments which have assisted in improving the manuscript.

Major comment:

We have reorganised the flow of the paper extensively as suggested and think this is much clearer and easier to follow.

Minor comments:

We have clarified the objectives (end of introduction, line 36-39) and methodology (line 40 - 48)

A paragraph on financial resources is included as point 4 amongst the identified challenges.

We have attempted to be clear where challenges differ between regions, for example we have included the following comment in line 141:   "With the exception of North Africa, this has not been robustly collected in the majority of African populations"

Reviewer 2 Report

The manuscript is well written. 

The meaning of abbreviations should be clarified the first time they appear in the text (for example ATS / ERS line 83/84)

There are some typographical errors in the text (for example, Dsease on line 156, or startegies on line 140)

Author Response

Reviewer 2:

Thank you for your review.

Response:

We have now clarified abbreviation ATS and ERS in line 110 and have corrected the typographical errors as suggested. Thank you for pointing these out.

Reviewer 3 Report

This paper is well written and adresses a very interesting topic. Spirometry in Africa is quite a neglected topic.

However the paper is written in a quite unusual form. It is not written in the form of a research paper, but the facts are also not presented in the form of a review. So it is not very clear what the  exact purpose of the paper is.

I advice to rewrite the paper so that the content can be presented in the form of a review. This would include also a methods section and a research question/hypothesis should be presented in the paper. 

Some minor comments:

Line 93: typo in the word "Operational"

Line 122: please specify which differences were found within different ethnicities

Line 152-179: contains suggestions for further research questions and improvement suggestions. This should rather be presented at the end of the paper.

Author Response

Reviewer 3:

Response:

Thank you for your helpful comments and suggestions which have strengthened this paper.

Major comments:

We have clarified the objectives (end of introduction, line 36-39) and methodology (line 40 - 48). In addition we have reorganised the flow of the paper extensively as suggested (e.g. moving research suggestions and improvement strategies moved to the end) and think this is now much clearer and easier to follow.

Minor comment:

The typographical errors identified have been corrected. 

The different ethnicities have now been clarified, line 145-147:   The Global Lung Initiative 2012 (GLI2012) has more recently provided the most robust and physiologically sound reference equations for multiple ethnicities (Caucasian, African American, North and South East Asian) for use in spirometry. 

Round 2

Reviewer 1 Report

I readed the revised version of the manuscript with interest and find that the authors took into account most of my concerns. However the grouping of thematics is still some difficult to follow.

I think that the manuscript  might be satisfactory for readers.

Author Response

Thank you for your comments and hope the flow of the paper is now clearer for the readers.

Reviewer 3 Report

The structure of the paper has improved significantly.

Just one suggestion for the methods: Line 44

" An internet search was also undertaken to look for unpublished information on spirometry training courses".

Please describe more precise. What search terms were used? Which search engines?

Author Response

Thank you for your comments.

We have added the following to methodology (line 44-45):    An internet search, using search engine Google Chrome with search terms “spirometry training” OR “lung function training” AND “Africa” or “South Africa”, was also undertaken to look for unpublished information on spirometry training courses.